# Is ChatGPT 3.5 smarter than Otolaryngology trainees? A comparison study of board style exam questions

**Jaimin Patel**[1], **Peyton Robinson**[2], **Elisa Illing**[1], **Benjamin Anthony**[1]*

**1** Department of Otolaryngology–Head and Neck Surgery, Indiana University School of Medicine, Indianapolis, IN, United States of America, **2** Indiana University School of Medicine, Indianapolis, IN, United States of America

* bpanthon@iu.edu

## Abstract

### Objectives

This study compares the performance of the artificial intelligence (AI) platform Chat Generative Pre-Trained Transformer (ChatGPT) to Otolaryngology trainees on board-style exam questions.

### Methods

We administered a set of 30 Otolaryngology board-style questions to medical students (MS) and Otolaryngology residents (OR). 31 MSs and 17 ORs completed the questionnaire. The same test was administered to ChatGPT version 3.5, five times. Comparisons of performance were achieved using a one-way ANOVA with Tukey Post Hoc test, along with a regression analysis to explore the relationship between education level and performance.

### Results

The average scores increased each year from MS1 to PGY5. A one-way ANOVA revealed that ChatGPT outperformed trainee years MS1, MS2, and MS3 (p = <0.001, 0.003, and 0.019, respectively). PGY4 and PGY5 otolaryngology residents outperformed ChatGPT (p = 0.033 and 0.002, respectively). For years MS4, PGY1, PGY2, and PGY3 there was no statistical difference between trainee scores and ChatGPT (p = .104, .996, and 1.000, respectively).

### Conclusion

ChatGPT can outperform lower-level medical trainees on Otolaryngology board-style exam but still lacks the ability to outperform higher-level trainees. These questions primarily test rote memorization of medical facts; in contrast, the art of practicing medicine is predicated on the synthesis of complex presentations of disease and multilayered application of knowledge of the healing process. Given that upper-level trainees outperform ChatGPT, it is

**Data Availability Statement:** All relevant data are within the manuscript and its Supporting Information files.

**Funding:** The author(s) received no specific funding for this work.

**Competing interests:** The authors have declared that no competing interests exist.

unlikely that ChatGPT, in its current form will provide significant clinical utility over an Otolaryngologist.

## Introduction

Current developments in artificial intelligence (AI) technology using advanced language models have generated a significant amount of public interest. Chat Generative Pre-Trained Transformer (ChatGPT), an AI-based language model developed by OpenAI, stands out for its ability to generate human-like responses in written format. Recent improvements to ChatGPT have garnered significant attention as this sophisticated AI platform finds its place in modern society. Fueled by vast databases, ChatGPT provides precise, personalized answers, a testament to its prowess in understanding the intricacies of human language. Based on this repository of knowledge, this language model effortlessly mirrors real-life conversations and boasts profound knowledge across diverse subjects [1].

The role of AI in medicine has been met with both hopeful intrigue as well as skepticism. AI-powered systems like ChatGPT can provide immediate access to information for patients and healthcare providers to augment healthcare decisions. ChatGPT seems to have an obvious role in patient education and medical education due to its ability to generate knowledgeable responses to fact-based questions with categorical answers. ChatGPT could possibly even play a direct role in augmenting patient care decisions and treatment. However, the accuracy and reliability of AI systems like ChatGPT has not yet been firmly established in medicine. Nevertheless, efforts continue to further develop this technology to determine if it holds value for patient care.

ChatGPT has been tested with a diverse list of standardized examinations, such as the uniform Bar Examination, the Scholastic Assessment test (SAT), the Graduate Record Examination (GRE), high school advanced placement exams and more [2]. Despite medicine being filled with niche terminology, acronyms, and multidisciplinary topics, ChatGPT has been able to exhibit a broad knowledge of medicine. Indeed, ChatGPT was found to likely be able to pass the USMLE Step 1 examination [3]. With regards to subspecialty fields, the literature has shown that ChatGPT is passable or near passable in board exams for Ophthalmology, Pathology, Neurosurgery, Cardiology, and Otolaryngology [3–9]; however, ChatGPT did quite poorly on the multiple-choice Orthopedic board exam [10]. As a repository of advanced medical knowledge, ChatGPT underperformed in comparison to the widely used UpToDate medical reference [11]. AI-based language models could be a great tool when patients desire reliable information on upcoming procedures, information on prescriptions, and other aspects of their care that carry significant weight to the patient [12], but their utility in advanced medical decision making remains to be investigated.

This current project compares the performance of ChatGPT version 3.5 to medical trainees at a US medical school and residency on board style questions for the Otolaryngology–Head and Neck Surgery board exam. To objectively quantify ChatGPT's knowledge of otolaryngology, we compared it to the infancy of medical education to senior level otolaryngology residents. The spectrum of questions ranged from fundamental concepts learned during the early years of medical school to the complexities of advanced medical and surgical patient management derived by the end of resident training. Our primary aim is to assess if and when ChatGPT can outperform human learners on Otolaryngology board style questions.

## Materials and methods

This study was exempt from requiring approval by the institutional review board at Indiana University. The study started collecting data on October 2nd, 2023, through January 5th, 2024. 30 multiple choice Otolaryngology board-style questions were asked to all years of medical students and Otolaryngology residents. The same questions were also asked to ChatGPT. Given that ChatGPT is a reiterative, learning-based model with a potential for different answers each time a question is asked, the test was administered to ChatGPT five times. These 30 questions were randomly aggregated with varying degrees of difficulty from pre-published board prep question bank with slight changes to the question-and-answer choices to prevent infringement of data. The questions are provided in supplement for review. The stem of the question and concepts were not changed to mimic the rigor of a board exam. Neither the human participants nor ChatGPT were asked to explain why they chose their respective answer to a question. However, ChatGPT did provide a reasoning to its choice without being prompted.

Questions were dispersed by using Google Forms to all 1461 medical students via listserv email, years 1–4, (MS1-MS4) and 17 of the 18 Otolaryngology residents, years 1–5, (PGY1-PGY5) at Indiana University School of Medicine. Participants were blinded to the purpose of this exam to avoid bias; thus, they were not provided informed consent on the underlying purpose of the study. They were simply asked to answer questions to test the quality of the questions written. No compensation or incentives were provided for the completion of this questionnaire. The only identifying data collected was the education level of each participant (MS1-PGY5). At the beginning of the study, the participants were given clear instructions: "Thanks so much for taking the time to answer this 30-question quiz that covers topics within Otolaryngology. We ask that you take this quiz in one sitting and do not use outside resources. This will allow us to accurately evaluate the questions written."

For ChatGPT, the model was prompted with the following: "You are a medical professional and I want you to pick an answer from the multiple-choice question I provide." For example, in one administration, ChatGPT responded with: "Of course, I would be happy to help you with multiple choice questions related to medical topics. Please provide the question and its options, and I'II do my best to provide you with the correct answer and explanation." Following this prompt, each of the 30 questions were provided one at a time. The answer and reasoning were recorded. The test was administered five times, once each day on five different days. This methodology was utilized to help capture the variability that language models can exhibit. We believe this allowed ChatGPT additional chances to retrieve the correct information within the vast databases it utilizes. Additionally, while ChatGPT was not solicited for an explanation, its reasonings were recorded for each response; however, we did not further analyze this data as it was not the intention of this study.

### Participants

The 30-question survey was completed by medical students and Otolaryngology residents at Indiana University (n = 48) and ChatGPT model 3.5 (n = 5). There were 9 education level groups across the human participants, MS1 (n = 8), MS2 (n = 7), MS3 (n = 10), MS4 (n = 6), PGY1 (n = 4), PGY2 (n = 4), PGY3 (n = 4), PGY4 (n = 2), and PGY5 (n = 3). See Table 1.

### Statistical analysis

Statistical analysis was conducted using Statistical Package for the Social Sciences (SPSS) [IBM]. A one-way ANOVA was conducted to compare Otolaryngology Board Exam Scores between human participants at each medical education level and ChatGPT. The ANOVA was implemented to identify if group differences were present between the 9 education levels

**Table 1. Demographics of participants.**

| Level of Education | Number of participants |
|---|---|
| MS 1 | 8 |
| MS 2 | 7 |
| MS 3 | 10 |
| MS 4 | 6 |
| PGY– 1 | 4 |
| PGY– 2 | 4 |
| PGY– 3 | 4 |
| PGY– 4 | 2 |
| PGY– 5 | 3 |
| MS–Medical Student Year, PGY–Post graduate year | |

(MS1-PGY5) and ChatGPT. Tukey's Honest Significant Difference Test (HSD) post hoc test was utilized to identify which of the 9 education levels (MS1-PGY5) differed to ChatGPT. A regression analysis was conducted to explore the relationship between education level and score, specifically to explore whether education level predicted score.

## Results

A regression revealed that the education level significantly predicted score $R^2$ = .765, $F_{(1, 46)}$ = 150.003, p < .001. The average score of human participants increased linearly as education level increased by years (MS1-PGY5) (MS1 = 28.75%; MS2 = 31.44%; MS3 = 36%; MS4 = 37.77%; PGY1 = 49.18%; PGY2 = 56.68%; PGY3 = 70.83%; PGY4 = 81.65%; PGY5 = 84.47%,). See Table 2.

The average score of ChatGPT was 54.66% across the 5 administrations. At times, ChatGPT did provide different answers to questions with different explanations. However, there was not a consistent increase in percent correct overtime. By mean, ChatGPT out-performed human participants from education level MS1-PGY1 but underperformed in comparison to PGY2-PGY5. See Fig 1.

A one-way ANOVA revealed that there were statistically significant differences in the average score between at least two of the 10 groups ($F_{(9, 43)}$ = [20.393], p < .001).

Tukey's HSD test for multiple comparisons were implemented to identify which groups differed significantly from each other, particularly from ChatGPT. Results revealed that the score

**Table 2. Percent correct and mean difference between ChatGPT and medical trainees.**

| Group A | Average % Correct | Group B | Average % Correct | Mean Difference (A-B) | Sig. | 95% Confidence Interval | |
|---|---|---|---|---|---|---|---|
| | | | | | | Lower Bound | Upper Bound |
| ChatGPT | 54.66 | MS1 | 28.75 | 25.91* | < .001 | 8.40 | 43.43 |
| | | MS2 | 31.44 | 23.22* | .003 | 5.22 | 41.21 |
| | | MS3 | 36.00 | 18.66* | .019 | 1.83 | 35.49 |
| | | MS4 | 37.77 | 16.89 | .104 | -1.72 | 35.50 |
| | | PGY-1 | 49.18 | 5.49 | .996 | -15.13 | 26.10 |
| | | PGY-2 | 56.68 | -2.01 | 1.000 | -22.63 | 18.60 |
| | | PGY-3 | 70.83 | -16.17 | .242 | -36.78 | 4.45 |
| | | PGY-4 | 81.65 | -26.99* | .033 | -52.70 | -1.28 |
| | | PGY-5 | 84.47 | -29.81* | .002 | -52.25 | -7.36 |
| MS–Medical Student Year, PGY–Post graduate year | | | | | | | |

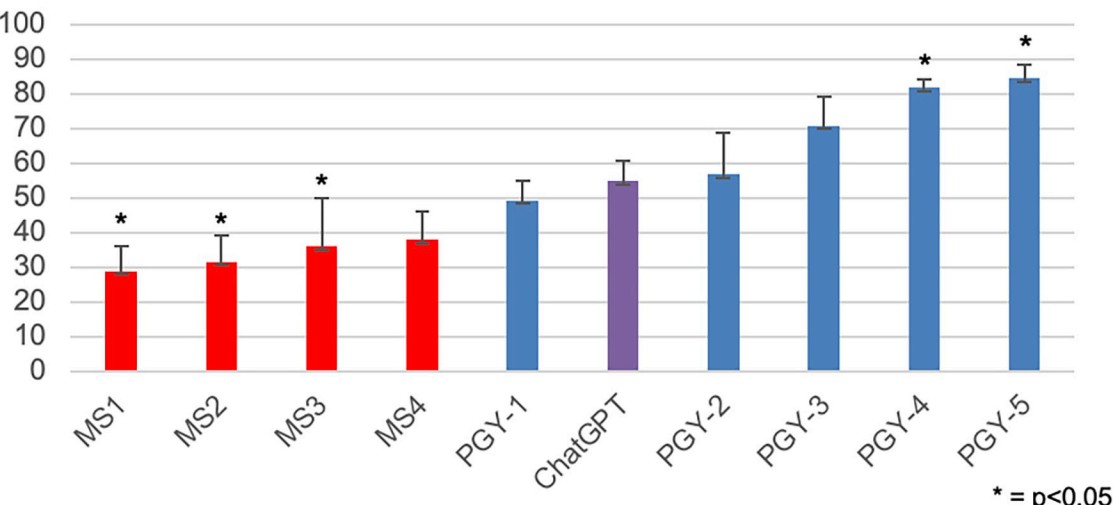

**Fig 1. Board exam scores between medical trainees and ChatGPT.**

significantly differed between ChatGPT and MS1 (p < .001, 95% C.I. = 8.3905, 43.4295), MS2 (p = .003, 95% C.I. = 5.2228, 41.2115), MS3 (p = .019, 95% C.I. = 1.8278, 35.4922), PGY-4 (p = .033, 95% C.I. = -52.7016, -1.2784), PGY-5 (p = .002, 95% C.I. = -52.2496, -7.3637).

Results revealed that the score did not significantly differ between ChatGPT and MS4 (p = .104, 95% C.I. = -1.7154, 35.5020), nor between ChatGPT and PGY-1 (p = .996, 95% C.I. = -15.1302, 26.1002), nor PGY-2 (p = 1.000, 95% C.I. = -22.6302, 18.6002), nor PGY-3 (p = .242, 95% C.I. = -36.7802, 4.4502).

## Discussion

Language-centric AI models, exemplified by ChatGPT, are gaining momentum for their ability to sustain coherent conversations as well as demonstrating aptitude on standardized examinations. Powered by deep machine learning techniques and extensive textual data, ChatGPT iteratively enhances its abilities via user interactions and reinforcement learning [1].

This study elucidates the comparison of otolaryngology knowledge of ChatGPT 3.5 to medical trainees. Findings reveal ChatGPT's superiority over beginners but eventual inferiority to seasoned tolaryngology residents on board-style questions targeting otolaryngology knowledge, indicating a progressive convergence in performance. Our senior residents scoring 85% tracks with historical data demonstrating that the written otolaryngology board exam has a 97% pass rate for senior residents scoring in the top 3 quartiles on their in-service exams [13]. Thus, this highlights that our questions align with the likely rigor of a board exam.

An additional key finding that we believe challenged ChatGPT was the nuanced and context-dependent nature of medical questions. Medical learners exhibited marked growth in their knowledge base, showcasing a linear progression in their average correct responses on the exam over years of continued training. This aligns with our expectations, as their evolving domain-specific knowledge, clinical experiences, and ability to interpret complex scenarios

increase with seniority. Human participants are adept at synthesizing information, applying critical thinking skills, and adapting responses to the intricacies of each scenario. This foundational skill is nurtured throughout the educational journey, particularly for individuals in the medical field. Resultantly, senior Otolaryngology residents demonstrate superior deductive abilities in answering multiple-choice questions compared to ChatGPT.

This AI model continues to struggle with advanced otolaryngology topics that require a deep understanding of current medical literature to properly navigate [14, 15]. This may be in part due to its lack of a deep understanding of patient-specific factors, consideration of evolving clinical contexts, and the incorporation of the latest medical research, specifically in Otolaryngology [11]. Future research should explore how AI language models can be trained to better perform answering medical queries. Further investigation should continue to be done to test the growth of ChatGPT as the model advances.

Albeit, the explanations for selecting an answer were unsolicited from ChatGPT and not a part of our intend study, there were instances where we noticed that ChatGPT seemed to grapple with a lack of understanding or data support, leading to what appeared as a guess, misinformed, or ill-informed answer. This was seen through multiple repetitions of the question with either similar answer choices but different explanations and vice versa. This has been demonstrated in multiple other study where ChatGPT struggled with the intricacies of medical knowledge resulting in its subpar performance [16, 17]. Overall, while illustrating the robust power of this language model, these inconsistencies beg the questions about continued knowledge gaps in specific queries on AI language models. Thus, while the model demonstrated an impressive ability to generate human-like responses in natural language, it continues to struggle with the intricacies and subtleties inherent in otolaryngology, and perhaps medicine generally. While this was an incidental finding in our study it would be a great opportunity for further research into ChatGPT's understanding of medical topics.

One limitation of this study was the small number of participants in the medical student group. While there was significance found in the comparison of the groups, there were still many students who did not answer the survey. This was most likely due to the inability to individually reach out to the large number of medical students at the university without creating bias among students. Also, the robust amount of information that is communicated to medical students could make the invitation to participate in this survey be missed. Additionally, the number of otolaryngology residents were also limited. To bolster a future study, we would recommend exploring the performance of trainees from multiple institutions.

## Conclusion

In conclusion, our findings emphasize the need for caution and meticulous assessment when deploying language models in specialized fields like otolaryngology or medicine, where precision is critical, and the stakes are high. ChatGPT showcases remarkable capabilities in natural language understanding and has been shown to pass a host of different board examinations [2–8]. In our study, ChatGPT scored an average of 54.66% which is similar to the 57% correct seen in Hoch et al. [9]. Considering this, ChatGPT is not yet intelligent enough to become the trusted gold standard to accessing medical information within Otolaryngology.

Additionally future research should focus on refining and tailoring language models for specific domains, incorporating real-time learning mechanisms, and addressing the interpretability challenges associated with automated systems in complex decision-making processes within the medical field. Consequently, with time, AI language models may evolve into indispensable tools for medical professionals and potentially even to patients and future research must aim to keep our understanding of their limits and abilities up to date.

## Supporting information

**S1 Dataset.**
(XLSX)

**S1 File.**
(DOCX)

## Author Contributions

**Conceptualization:** Jaimin Patel, Peyton Robinson, Benjamin Anthony.

**Data curation:** Jaimin Patel, Peyton Robinson.

**Formal analysis:** Jaimin Patel.

**Investigation:** Jaimin Patel.

**Methodology:** Jaimin Patel.

**Project administration:** Jaimin Patel.

**Software:** Jaimin Patel.

**Supervision:** Elisa Illing, Benjamin Anthony.

**Visualization:** Jaimin Patel, Peyton Robinson.

**Writing – original draft:** Jaimin Patel, Peyton Robinson, Benjamin Anthony.

**Writing – review & editing:** Jaimin Patel, Peyton Robinson, Benjamin Anthony.

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
