## [Decision Letter · Decision Letter 0]

4 Jul 2024

PONE-D-24-22688Is ChatGPT smarter than Otolaryngology trainees? 

A comparison study of board style exam questionsPLOS ONE

Dear Dr. Patel,

Thank you for submitting your manuscript to PLOS ONE. After careful consideration, we feel that it has merit but does not fully meet PLOS ONE’s publication criteria as it currently stands. Therefore, we invite you to submit a revised version of the manuscript that addresses the points raised during the review process.

We look forward to receiving your revised manuscript.

Kind regards,

Harpreet Singh Grewal

Academic Editor

PLOS ONE

Journal Requirements:

2. We note that your Data Availability Statement is currently as follows: All relevant data are within the manuscript and its Supporting Information files

Additional Editor Comments:

Well conceptualized manuscript. However, the selection criteria for the questions has to be clarified. Did they represent the old ENT board exam questions ? Why were Medical students included and not only ENT trainees since only they will potentially take the board exams and not the med students. There were also some issues with discussion and introduction section. Following are some hard recommendations and some suggestions for improvement:

HARD RECOMMENDATIONS:

-Please provide selection criteria for the questions. How were they selected ? Were they graded ? Did they represent some of the old ENT board exam question ?

-Please explain why med students were included in the cohort since they do no typically take the board exam, which is potentially only to be taken by the residents/fellows ? This sets different playing fields for ChatGPT which could have skewed the results.

-Please touch upon why the ChatGPT was asked to provide additional reasoning for a question where as the human participants were just answering the question. This may have altered the results. Please address the rationale behind this.

-The discussion section needs to be more nuanced, by referring to studies that have already done something similar and then comparing your results against those. Please note that discussion section should always start by summarizing the results, then compare the studies results with some of the similar previous work that has been done. Then add strengths and limitations section, followed by a conclusion. The comparison narrative on your discussion seems weaka nd should benefit by referencing more studies that have done some similar work previously.

SUGGESTIONS FOR IMPROVEMENT:

-Please shorten the discussion section. Some of the studies mentioned there should actually be moved to your discussion section. This would help rooting your study in the prior literature, as i have mentioned above.

Reviewers' comments:

Reviewer's Responses to Questions

**Comments to the Author**

1. Is the manuscript technically sound, and do the data support the conclusions?

Reviewer #1: Yes

Reviewer #2: Yes

Reviewer #3: No

2. Has the statistical analysis been performed appropriately and rigorously? 

Reviewer #1: Yes

Reviewer #2: Yes

Reviewer #3: Yes

3. Have the authors made all data underlying the findings in their manuscript fully available?

Reviewer #1: Yes

Reviewer #2: No

Reviewer #3: No

4. Is the manuscript presented in an intelligible fashion and written in standard English?

Reviewer #1: Yes

Reviewer #2: Yes

Reviewer #3: Yes

5. Review Comments to the Author

Reviewer #1: The article compared the performance of ChatGPT with that of human. Simple and straightforward study. Just need to cite some more recent literature in the discussion like https://ijvlms.sums.ac.ir/article_49682.html. The version of the chatGPT can also be mentioned in the title of the article so that readers understand at a glance which chatgpt you are referring to.

Reviewer #2: Thank you for this interesting paper. While the results no doubt add to our understanding of AI in medicine, there are a few comments to be made:

1) You do not provide any information about the questions asked of the two groups. How were they constructed? Are the questions at different levels or graded, and how do we know that? The quality of the questions and answers no doubt have implication on the results and should be shared with the reader. This is particularly given your conclusion that ChatGPT struggles with harder questions, but we don't have the data to know that's where it struggled or that's where higher learners didn't struggle. Also, do we have a comparison to know that higher learners do this well on these tests in general? Is a score of 85% expected from PGY5s?

2) The article suggests you asked ChatGPT to provide an answer and then explain their reasoning. It does not appear this was done for human participants. Why is that? Your article suggests there were times that the answers and the explanations did not match. Could this also not be the case for humans? Again, also knowing the questions could provide insight here. Were there certain questions that were more likely to trip up humans or ChatGPT?

3) What is the denominator of participants? How many people were invited to participate? Is there potential bias introduced based on who participates?

4) Please include potential issues with this study in your discussion.

5) Your discussion includes a paragraph about the ethical implications of AI. It does seems to fit with the paper itself (ie the study doesn't focus on AI ethics) nor are there any references provided. You may want to remove this, and from the conclusion.

Reviewer #3: This is a well written but not rigorously conducted study on the test taking abilities of chatgpt3.5. The strength of the paper Include its simply and coherently written introduction and methods. The weakness of this paper is in the rigor of the experiment design and the construction of the test for chatgpt. Firstly, it is unclear whether the 30 questions chosen are valid representations of otolaryngology board examination questions. There is no explanation of how they were chose, how they were vetted and against what criteria, and what mix of topics/difficulty they test. Many articles have been written on similar topics that can be referenced for their methodology. (Ideally, the questions used would be real board questions, but in the absence of that, there are still ways to objectively vet and classify questions. For example, see https://pubs.rsna.org/doi/10.1148/radiol.230582).

Secondly, the discussion does not root the paper within the existing literature, with zero citations or references to other similar work. The discussion actually begins to present new information (anecdotal observations of how chatgpt is reasoning through questions or not) that was not shown in results

Lastly, the authors should try alternative prompts to see if chatgpt may Perform better if, for example, it was told it is a practicing otolaryngologist rather than a medical professional, etc.

6. PLOS authors have the option to publish the peer review history of their article (what does this mean?). If published, this will include your full peer review and any attached files.

Reviewer #1: No

Reviewer #2: No

Reviewer #3: No

---

## [Author Response · Author response to Decision Letter 0]

28 Aug 2024

The response is included as a separate file for review.

---

## [Editor Report · Decision Letter 1]

4 Sep 2024

Is ChatGPT 3.5 smarter than Otolaryngology trainees? 

A comparison study of board style exam questions

PONE-D-24-22688R1

Dear Dr. Patel,

We’re pleased to inform you that your manuscript has been judged scientifically suitable for publication and will be formally accepted for publication once it meets all outstanding technical requirements.

Kind regards,

Harpreet Singh Grewal

Academic Editor

PLOS ONE
---

## [Editor Report · Acceptance letter]

17 Sep 2024

PONE-D-24-22688R1 

PLOS ONE

Dear Dr. Patel, 

I'm pleased to inform you that your manuscript has been deemed suitable for publication in PLOS ONE. Congratulations! Your manuscript is now being handed over to our production team.

Kind regards, 

on behalf of

Dr. Harpreet Singh Grewal 

Academic Editor

PLOS ONE